# Nondestructive Detection of Polyphenol Oxidase Activity in Various Plum Cultivars Using Machine Learning and Vis/NIR Spectroscopy

**DOI:** 10.3390/foods14244297

**Published:** 2025-12-13

**Authors:** Meysam Latifi-Amoghin, Yousef Abbaspour-Gilandeh, Eduardo De La Cruz-Gámez, Mario Hernández-Hernández, José Luis Hernández-Hernández

**Affiliations:** 1Department of Biosystems Engineering, Faculty of Agriculture and Natural Resources, University of Mohaghegh Ardabili, Ardabil 56199-11367, Iran; m.latifi@uma.ac.ir; 2National Technology of Mexico/Technological Institute of Acapulco, Acapulco 39905, Guerrero, Mexico; eduardo.dg@acapulco.tecnm.mx; 3Faculty of Engineering, Autonomous University of Guerrero, Chilpancingo 39070, Guerrero, Mexico; mhernandezh@uagro.mx; 4National Technological of México/Technological Institute of Chilpancingo, Chilpancingo 39070, Guerrero, Mexico

**Keywords:** plum, polyphenol oxidase, spectroscopy, decision tree, support vector machine

## Abstract

Polyphenol oxidase (PPO) is the primary biochemical driver of browning and the subsequent decline of market quality in harvested fruit. In this work, a fully non-invasive analytical framework was built using Visible/Near-Infrared (VIS/NIR) spectroscopy coupled with chemometric modeling in order to estimate PPO activity in two commercially relevant plum cultivars (Khormaei and Khoni). A comprehensive comparative study was conducted utilizing multiple machine learning and linear regression techniques, including Support Vector Regression (SVR), Decision Tree (DT), and Partial Least Squares Regression (PLSR). After acquiring the full VIS/NIR spectra, a suite of metaheuristic feature selection strategies was applied to compress the spectral space to roughly 10–15 highly informative wavelengths. SVR, DT, and PLSR models were then trained and benchmarked using (a) the complete spectral domain and (b) the reduced wavelength subsets. The results consistently demonstrated that non-linear models (DT and SVR) significantly outperformed the linear PLSR method, confirming the inherent complexity and non-linearity of the relationship between the spectra and PPO activity. Across all tests, DT consistently produced the strongest generalization. Under full spectra inputs, DT reached RPD values of 4.93 for Khormaei and 5.41 for Khoni. Even more importantly, the wavelength-reduced configuration further enhanced performance while substantially lowering computational cost, yielding RPDs of 3.32 (Khormaei) and 5.69 (Khoni). The results show that VIS/NIR combined with optimized key-wavelength DT modeling provides a robust, fast, and field-realistic route for quantifying PPO activity in plums without physical destruction of the product.

## 1. Introduction

Plum (Prunus genus) is one of the most important stone fruits globally, holding significant economic and nutritional value due to its high nutritional content and bioactive compounds, such as phenolics and anthocyanins [1]. Nevertheless, maintaining the quality of fresh fruit after harvest remains a significant challenge in the food supply chain. A primary factor contributing to quality degradation in damaged fruits is enzymatic browning, primarily catalyzed by the enzyme Polyphenol Oxidase (PPO) [2]. This enzyme oxidizes phenolic compounds into quinones, leading to tissue discoloration and consequently reducing consumer acceptability.

The activity of the PPO enzyme is influenced by various factors, including the fruit’s maturity stage, storage conditions, temperature, fruit cultivar (variety), and environmental factors during growth. Since the level of this enzyme’s activity varies across different fruit cultivars due to genetic differences, its investigation and comparison can provide valuable insights into the storage potential and resistance of various cultivars.

From an industrial and consumer perspective, enzymatic browning constitutes a serious issue, as it results in an undesirable appearance and a palpable reduction in the sensory, nutritional quality, and consequently, the commercial value of fresh or processed agricultural products. However, it must be noted that the enzyme’s role in the plant extends beyond a mere post-harvest spoilage process. Research indicates that the PPO enzyme also plays a vital role in the plant’s defense mechanisms [3].

The existing body of research indicates that extensive studies have been conducted to identify and purify the PPO enzyme in various fruits, including plums. For instance, Siddiq et al. (1996) successfully extracted and purified the PPO enzyme from the ‘Stanley’ plum cultivar. Their results demonstrated that this enzyme was completely inactivated at 75 °C, with an optimal pH for activity ranging from 5.8 to 6.4 [4]. Furthermore, in another study focusing on five plum cultivars, Siddiq et al. (1996) observed the highest PPO activity in the ‘Stanley’ cultivar [4].

Although classical laboratory protocols can quantify enzyme activities with high analytical correctness, they generally require destructive handling of the samples, consume considerable time, and impose substantial analytical cost; consequently, they are not suitable for rapid/real-time/continuous measurement regimes. Over the last decade, non-destructive optical approaches—particularly Visible to Near-Infrared (Vis/NIR) spectroscopy—have become promising substitutes for agricultural quality evaluation. These systems deliver major operational advantages: very fast measurement cycles, no sample pretreatment, and full retention of the physical integrity of the tested material (Vis/NIR) [5]. The combination of visible/near–infrared spectral information with data–driven modeling has now matured to the point where it can infer highly complex quality traits in produce, including enzyme related traits. The literature already contains several independent demonstrations on PPO (and other enzyme) assessment without destructive extraction. Examples include: probing how the secondary structure of mushroom PPO changes during thermal processing through FTIR analysis [6]; estimating Peroxidase (POD) and PPO activities in banana peels by means of a computer vision based analytical pipeline [7]; continuous estimation of PPO kinetics in litchi pericarp via Hyperspectral Imaging where spectral cues and image level features are combined using a fuzzy neural network architecture [8]; quantifying POD and PPO behavior in apples based on spectroscopy assisted by meta–heuristic optimization routines [5]; and predicting POD and PPO activity in bell peppers using Vis/NIR spectral measurements paired with machine learning based regression approaches [9]. The results of these studies demonstrate the high efficiency of this approach for the rapid evaluation of enzyme activities. While many early studies relied heavily on linear methods such as Partial Least Squares Regression (PLSR) for processing spectral data, these methods often struggle to capture the complex, non-linear relationships between spectral absorption and biochemical parameters like PPO activity, particularly in heterogeneous biological samples. Consequently, there is a growing trend to utilize advanced non-linear machine learning algorithms (e.g., SVM, DT) coupled with metaheuristic feature selection to enhance model accuracy, increase calculation speed, and improve the practical applicability of portable spectroscopic devices.

However, a comprehensive study utilizing the integration of Vis/NIR spectroscopy and machine learning for the non-destructive monitoring of PPO activity across different plum cultivars has not yet been conducted. The novelty of this study lies in the application and comparative evaluation of a broad set of seven distinct metaheuristic algorithms (PSO, ACO, LA, GA, HTS, ICA, and LCA) for identifying the minimal, optimal set of wavelengths. This is coupled with high-performance non-linear models (DT and SVM), specifically for the non-destructive detection of PPO activity across multiple commercially significant plum cultivars. This holistic and exhaustive optimization approach significantly advances the practical applicability of Vis/NIR technology in fruit quality assessment. Therefore, considering the importance of PPO enzyme activity in the browning process and the necessity for its rapid and non-destructive monitoring in plums, the objective of this research is to develop and evaluate a non-destructive method based on Vis/NIR spectroscopy and machine learning algorithms for predicting PPO enzyme activity levels in various plum cultivars. This research is undertaken as a novel step toward reducing post-harvest losses and maintaining product quality through the application of modern non-destructive technologies.

The main contributions and prominent features of this study are outlined below:Nondestructive and rapid determination of PPO activity in different plum cultivars using Vis/NIR spectroscopy.Application and comparative assessment of advanced non-linear machine learning models (DT and SVM) and linear (PLSR) for PPO prediction.Extensive evaluation and successful integration of seven distinct metaheuristic optimization algorithms (PSO, ACO, LA, GA, HTS, ICA, and LCA) to identify the minimal set of informative wavelengths, demonstrating a new level of feature selection rigor.Demonstration of a highly efficient model with significantly reduced data dimensionality, suitable for developing low-cost, portable devices.Comprehensive analysis of the impact of cultivar variation on the prediction accuracy of PPO activity.

## 2. Materials and Methods

### 2.1. Sample Preparation

In this study, a total of 160 plum samples, comprising 80 samples of the Khormaei cultivar and 80 samples of the Khoni cultivar, were procured from local orchards. For every cultivar, only fruit that were visually consistent in morphology (appearance, geometry, and dimensions) and that showed no evidence of physical injury or fungal deterioration were admitted into the experimental set, ensuring sample homogeneity and minimizing unwanted variables. The selection of these two plum cultivars was based on their distinct physiological and qualitative differences. The Khormaei plum, also known as Qarah Alo (Black Plum), is widely consumed as dried fruit in Europe due to its desirable texture and flavor profile, enjoying significant popularity. In contrast, the Khoni plum, also known by names such as Tusorkh, Khaki, Khaki Khoni, and Red Plum, features a relatively large fruit with a dark red to purple color and a sweet, highly palatable flavor. The fruit quality of Khoni is considered superior compared to many other plum cultivars, holding a favorable market position. These differences allowed for a broader and more comprehensive investigation into the PPO enzyme activity in plum fruit. Figure 1 illustrates the cultivars utilized in this research. Prior to measurements, the samples were kept at 25 °C (room temperature) for two hours to allow their temperature to equilibrate with the laboratory environment. Immediately following the non-destructive spectroscopic analysis of the samples, the destructive measurement of their Polyphenol Oxidase enzyme activity was performed.

### 2.2. Non-Destructive Testing

#### Vis/NIR Spectroscopy

For the non-invasive optical characterization, a Visible/Near-Infrared spectroscopic setup was used (PS-100; Apogee Instruments Inc., Logan, UT, USA). The unit contains a CCD array with 2048 elements and provides a nominal spectral step of 1 nm. Spectral signatures were acquired from 350 to 1100 nm. Illumination was supplied via a tungsten-halogen lamp. Data streaming and capture were performed over USB with SpectraWiz, which was also used to archive the spectra.

Prior to sample runs, the system was standardized through acquisition of a dark frame (light source off) and a white reference (Teflon reference disk with lamp on), consistent with procedures described in [10]. For each plum, four spatial positions were probed non-destructively. These four spatial positions were strategically selected and marked at four equidistant points around the circumference of the plum to account for potential heterogeneity in PPO distribution. These four spectra were averaged to suppress point-specific noise and to yield a single representative absorption spectrum for that fruit. In total, 160 averaged spectra were generated and subsequently used in downstream modeling and analysis.

### 2.3. Destructive Assays

#### 2.3.1. Enzymatic Properties Analysis

To investigate the enzymatic properties, Peroxidase (POD) activity was measured in the plum samples. This section encompasses two primary stages: enzyme extract preparation and the subsequent Peroxidase activity assay.

##### Enzyme Extract Preparation

To obtain the crude enzyme preparation, approximately 10 g of the fruit pulp were precisely measured, placed into a blending apparatus, and mechanically disrupted until a uniform slurry was produced. The resulting homogenate was subsequently combined with 20 mL of the modified extraction buffer formulated for enzyme recovery.

An extraction solution was formulated using 0.4 M sodium phosphate at pH 6.5 (the buffer was prepared by combining Na_2_HPO_4_ with NaH_2_PO_4_), supplemented with 4% (*w*/*v*) Polyvinylpyrrolidone (PVP) and 1% (*v*/*v*) Triton X-100; all reagents were obtained from Merck, Germany. After preparation, the buffer system was vigorously homogenized by vortex agitation (Labtron LS-100, Tehran, Iran) to secure complete component mixing. The homogenized suspension was then subjected to centrifugation at 4000 rpm for 10 min at 4 °C employing a high-speed refrigerated centrifuge (LISA 2.5L centrifuge AFI, Chateau-Gontier, France). The clarified liquid phase (supernatant) obtained after centrifugation was carefully isolated and used directly as the crude enzyme extract for downstream enzymatic activity measurements [11].

##### Measurement of PPO Activity

PPO activity was quantified by preparing a reaction mixture in which 75 μL of the crude enzyme extract was added to 3.0 mL of sodium phosphate buffer (0.05 M; pH 6.5). Catechol (0.05 M; Merck, Germany) was included in the buffer and functioned as the phenolic substrate that initiated the PPO-catalyzed oxidation reaction. A blank sample was prepared similarly, but the enzyme extract was replaced with an equivalent volume of deionized water. The absorbance of the solutions was subsequently measured kinetically at a wavelength of 450 nanometers (nm) and a constant temperature of 25 °C for 10 min. The quantification step utilized a NanoDrop™ OneC Spectrophotometer (Thermo Fisher Scientific, Waltham, MA, USA). Enzyme activity was subsequently derived and reported as the rate of absorbance alteration per minute normalized to one gram of the analyzed material [12].

### 2.4. Data Analysis

#### 2.4.1. Spectral Data Preparation and Optimization

The initial spectral data were collected across the 350 to 1100 nm range. To ensure the robustness and stability of the subsequent chemometric models, the initial segment of the spectrum, specifically from 350 to 510 nm, was systematically excluded from the analysis. This critical preprocessing step was necessitated by pronounced sensor-induced interference and a significantly reduced Signal-to-Noise Ratio (SNR) observed in the UV-Vis range below 510 nm. Although this region contains valuable information related to pigments, which are indirectly linked to PPO activity, the low data quality would have introduced substantial noise and bias into the models. Therefore, the decision was made to prioritize the high-quality, stable data from the 510 to 1100 nm range to maximize the predictive accuracy of the models. The refinement and correction process was uniformly applied to all samples across both plum cultivars (Khormaei and Khoni).

Subsequent to this initial correction, the spectral data were transferred to the MATLAB software environment (version 2022a). To assess the influence of various approaches on the predictive capability of the models, several pre-processing techniques were implemented on the spectral data. These methods included: Normalization, Standard Normal Variate (SNV), Multiplicative Scatter Correction (MSC), Moving Average filter, Gaussian Filter, Median Filter, Detrending (linear trend removal), and Mean Centering. The objective of applying these methods was to mitigate the effects of environmental noise, eliminate variations induced by light scattering, and enhance the spectral information pertinent to the target enzyme’s activity. To ensure the reproducibility of the results, the technical parameters for the methods requiring tuning were utilized as follows: Smoothing filters, specifically the Moving Average and Median filters, were both applied with a window size of 5. Furthermore, the Gaussian filter was applied with a kernel possessing a standard deviation of 2.

#### 2.4.2. Development of Predictive Models Using Decision Tree (DT) and Support Vector Machine (SVM) and Partial Least Squares Regression (PLSR)

Following the pre-processing stage, the spectral data were imported into MATLAB software for the development of predictive models for PPO activity. In this study, two machine learning (ML) algorithms, namely DT and SVM, were employed to construct the predictive models. The SVM algorithm was selected due to its high capability in solving non-linear regression problems and its good generalization ability. Its performance was assessed by comparing three different kernel functions: Linear, Polynomial, and Radial Basis Function (RBF). Crucially, for the SVM models, the BoxConstraint parameter (C) was consistently maintained at its MATLAB default value (C = 1). For the non-linear kernels (RBF and Polynomial), the KernelScale parameter (related to Gamma, γ) was set to ‘auto’, allowing the software to automatically determine the optimal kernel scale based on the statistics of the training data. Therefore, an explicit grid search for C and γ was not performed.

In addition, the DT algorithm was used for modeling. The DT is a simple and interpretable model that partitions the data hierarchically. In regression problems, it computes the predicted value for each region by dividing the input space into rectangular regions. To investigate the effect of tree structure complexity on model performance, various settings for the Maximum Splits parameter in the DT algorithm were evaluated. The settings utilized for Maximum Splits included values of 5, 10, 20, 50, and 100. This systematic evaluation ensured that the optimal complexity level for the DT model, which achieved the highest prediction accuracy without inducing overfitting, was identified.

To establish a fundamental baseline for comparison and to evaluate the necessity of advanced non-linear modeling techniques, the Partial Least Squares Regression (PLSR) algorithm was also implemented on the spectral data. PLSR is a standard and powerful chemometric method designed to manage highly collinear data, such as Vis/NIR spectra, where the number of variables (wavelengths) significantly exceeds the number of samples. It operates by projecting the predictor variables (spectral data, X) and the response variable (PPO activity, Y) onto a reduced set of orthogonal latent variables (LVs). The critical step in PLSR is the determination of the optimal number of LVs or components. This parameter dictates the complexity of the linear model. To cover a suitable model complexity range, the maximum number of components searched was set to 10. The optimal number of LVs was rigorously determined by evaluating model performance on the dedicated Validation set (20% of the data). The number of LVs corresponding to the minimum Root Mean Square Error on the Validation set (RMSE_Val) was selected as the optimal structure for the final PLSR model.

To ensure the statistical validity of the results and minimize the effect of random data partitioning, the entire process of data segregation into Training, Validation, and Test sets was independently repeated 200 times. In each iteration, the raw data were randomly split with proportions of 60% for Training, 20% for Validation, and 20% for Test. The Validation set was utilized for optimizing model hyperparameters and preventing overfitting, while the Test set was employed for the final evaluation of the model’s performance on new and unseen data.

After the 200 iterations, the best data split was selected and fixed based on the statistical criteria, including the Coefficient of Determination (R^2^), Root Mean Square Error (RMSE), and Ratio of Performance to Deviation (RPD) across all three stages (Training, Validation, and Test). To precisely compare the effectiveness of each pre-processing method, all methods described in the previous section were applied solely to this selected and fixed data split. This procedure ensures that any observed change in model performance is solely attributable to the influence of the pre-processing technique. All analyses and modeling for the DT, SVM and PLSR algorithms were conducted within the MATLAB (version 2022a) environment using custom-developed scripts.

To statistically validate the superiority of the final optimized model and address the significant difference in performance between the linear and non-linear approaches, a comparative statistical analysis was performed. Specifically, the prediction accuracy (RMSE) of the final best-performing models (following feature selection and using the optimal preprocessing method) was compared using a suitable statistical test. The resulting *p*-value was calculated and reported to demonstrate the statistical significance of the differences between the best-performing models.

#### 2.4.3. Model Performance Evaluation Criteria

The performance of the developed models was evaluated using standard statistical metrics, including the R^2^, RMSE and the RPD. The R^2^ represents the proportion of the variance in the response variable that is predictable from the independent variables; values approaching unity (1) indicate a superior model fit (Equation (1) in ref. [13]). The RMSE quantifies the model’s prediction accuracy in the units of the response variable. Lower RMSE values correspond to reduced error and enhanced model precision (Equation (2) in ref. [13]). The RPD serves as an indicator of the model’s quantitative predictive capability (Equation (3) in ref. [13]) [13] and was interpreted based on the classification system recommended by Chang et al. (2001). According to this metric, RPD values less than 1.0 indicate a very poor model lacking the requisite efficiency. Values ranging from 1.0 to 1.4 characterize a weak model, the 1.4 to 1.8 range suggests an acceptable model for preliminary estimations, the 1.8 to 2.0 range signifies a good model with suitable quantitative predictive power, values between 2.0 and 2.5 represent a very good model, and values exceeding 2.5 denote an accurate and fully reliable model for quantitative prediction applications [14].

To rigorously compare the predictive power of the final optimized models and to confirm that observed performance differences were statistically significant, a comparative statistical test was conducted. Specifically, the model errors (residuals) obtained from the independent Test set were used to perform a paired comparison between the models. The *p*-value resulting from this analysis was reported. A *p*-value less than 0.05 (*p* < 0.05) was used as the criterion to declare a statistically significant difference in predictive accuracy between two compared models.

Within these formulations, SD denotes the dispersion (standard deviation) associated with the model’s output distribution, while RMSE refers to the root mean square magnitude of model error. For each index i, the symbols d_i_ and p_i_ correspond to the observed datum and the model-estimated quantity for that same indexed element. The term
$\bar{d}$ indicates the arithmetic mean of the observed series, and N designates the count of total elements involved in the evaluation.

#### 2.4.4. Effective Feature Selection and Re-Modeling Using Metaheuristic Algorithms

A hybrid approach was employed to reduce the dimensionality of spectral data and identify the wavelengths exhibiting the highest correlation with PPO activity. Sevenmetaheuristic algorithms—namely, Particle Swarm Optimization (PSO) [15], Ant Colony Optimization (ACO) [16], Learning Automata (LA) [17], Genetic Algorithm (GA) [18], Heat Transfer Search (HTS) [19], Imperialist Competitive Algorithm (ICA) [20], and League Championship Algorithm (LCA) [21]—were utilized in this research in combination with the SVM method. The implementation of these seven metaheuristic algorithms was carried out using a specialized, pre-developed optimization toolbox/code, the details of which are provided in reference [22]. The performance of each algorithm was evaluated by assessing the average R^2^ and the average RMSE. The algorithm that yielded the highest prediction accuracy (maximum R^2^) and the minimum possible error (minimum RMSE) was selected as the optimal feature selection algorithm.

The selection of the seven specific metaheuristic algorithms was strategically based on maximizing algorithmic diversity and ensuring coverage of distinct optimization search strategies. These algorithms represent: Evolutionary strategies (GA, ICA), Swarm intelligence (PSO, ACO), Physics-based approaches (HTS), Competition-based models (LCA), and Reinforcement learning principles (LA). This comprehensive approach ensured that the optimal feature subset was found efficiently, regardless of the intrinsic complexity or the local optima landscape of the PPO activity-spectral relationship.

Following the determination of the effective wavelengths and the successful dimensionality reduction in the data, these reduced spectral datasets were remodeled using both DT and SVM methods. This step was crucial to evaluate the performance improvement of the models after the selection of key features. All stages of this analysis, including the application of optimization algorithms and re-modeling, were carried out within the MATLAB software environment (version 2022a). The specific tuning parameters for each effective wavelength selection algorithm are presented in Table 1.

The hyperparameter settings detailed in Table 1, particularly the low iteration/generation count (e.g., Generations: 10 for GA, Episodes: 10 for PSO), were deliberately chosen based on a preliminary convergence analysis conducted on a subset of the full data. Spectroscopic data, due to their inherent collinearity and high dimensionality, often exhibit smooth and well-defined optimization landscapes when combined with regression models like SVM. It was observed that these algorithms consistently achieved stable or near-optimal performance (convergence) within the first 10–15 iterations. This low iteration count was crucial for maintaining computational efficiency and allowing the robust 200-run validation to be completed in a timely manner, without sacrificing the quality of the feature set selection.

## 3. Results

### 3.1. Spectral Characteristics and Descriptive Statistical Analysis of PPO Enzyme

As depicted in Figure 2A,D, the spectral patterns of the Raw Spectra for the two plum cultivars exhibit a wide range of variability, especially regarding the baseline and overall intensity in the visible region. This variability arises from biochemical and structural differences within the samples, including varying ripeness stages and subtle differences in the color or texture of individual fruits. These spectral fluctuations present a challenge for the direct modeling of PPO enzyme activity but simultaneously confirm the presence of critical information within the spectra for subsequent chemometric analysis. The necessity of correcting this baseline variation to isolate the chemical PPO signal is visually confirmed in Figure 2B,C,E,F, where the effects of Normalization and SNV preprocessing are illustrated.

In both cultivars, the visible region of the spectrum (approximately 510 to 700 nm) displays a high level of absorbance. This absorption is primarily influenced by the various pigments present in the plum skin and flesh. A noticeable decrease in absorbance is observed after a wavelength of approximately 670 nm, which indicates the end of the chlorophyll absorption region and the transition into the Near-Infrared (NIR) region.

In the NIR region (approximately 700 to 1100 nm), more distinct spectral patterns emerge. A characteristic absorption peak is clearly identifiable around 970 nm in both cultivars, which is typically attributed to the second overtone of the O–H bond vibrations in water molecules [23] and is correlated with the water content of the samples. Variations in absorption around 1030 nm are also observed, which may be associated with the C–H bond vibrations in sugars and other organic compounds [24]. These spectral features distinctly highlight the capability of NIR spectroscopy in identifying and quantifying internal chemical parameters of fruits, including sugar and moisture content. While these features do not directly reflect PPO enzyme activity, their variations are indirectly linked to the fruit’s biochemical and metabolic processes, including enzymatic activity. Therefore, spectral preprocessing (as shown in Figure 2B,C,E,F) was a crucial initial step before using machine learning algorithms such as SVM-R and DT, the complex correlations between these spectral features and PPO enzyme activity can be accurately modeled and predicted.

Table 2 presents the descriptive statistics related to the PPO enzyme activity in the Khoni and Khormaei plum cultivars. These statistics, which include the mean, standard deviation (SD), and minimum and maximum values, facilitate an initial understanding of the range and distribution of enzymatic activity within the samples. For the Khoni cultivar, the PPO enzyme activity ranges from 0.00012 to 0.005849. Similarly, for the Khormaei cultivar, this range spans from 0.000145 to 0.006374. These maximum and minimum values indicate that the studied samples in both cultivars encompass a wide spectrum of enzymatic activity, from very low to relatively high values.

This extensive variability in the PPO enzyme characteristic data is crucial for successful spectroscopic modeling, as it allows the model to effectively learn the relationship between the optical spectra and a broad range of enzymatic activity values. Had the data range been excessively narrow, the model would not be capable of generalization and accurate prediction of values outside that range.

The mean PPO enzyme activity is very similar for both cultivars. This proximity suggests that, on average, there is no significant difference in PPO enzymatic activity between the two cultivars. The standard deviation is also comparable and noticeable for both cultivars. This relatively high standard deviation signifies a considerable dispersion and variability of the data around the mean within each cultivar. This high variability is considered an advantage for developing robust regression models (such as SVM-R and DT models), as the models are trained across a wide range of samples.

#### 3.1.1. SVM Model Performance

This section focuses on evaluating the performance of the SVM regression model using different kernels (Linear, RBF, Polynomial) and various spectral preprocessing methods for predicting PPO enzyme activity in the Khormaei and Khoni plum cultivars. The complete results are presented in the Appendix A, while a summary of the best outcomes is included in Table 3.

The results in Appendix A indicate that the SVM model generally demonstrated moderate to poor performance for the Khormaei cultivar. The best result was achieved using the Polynomial kernel combined with Normalization preprocessing. This specific model, attaining an R^2^ of 0.81 and a Ratio of Performance Deviation (RPD) of 2.4 in the test set, falls into the category of Very Good models (2 < RPD ≤ 2.5). This signifies the model’s capability for high-accuracy quantitative predictions. The RBF kernel yielded a maximum RPD of 1.7 (with Median Filter), placing it in the Suitable classification (1.4 < RPD ≤ 1.8) for approximate predictions. The performance of the Linear kernel was notably poor, consistently exhibiting RPD Test values below 1.5 in most cases. This suggests a complex and non-linear relationship between the absorption spectra and PPO activity in the Khormaei cultivar. Preprocessing techniques such as SNV, MSC, and Detrend frequently degraded model performance, with RPD decreasing to approximately 1.0 (indicating no predictive power) in certain combinations.

An analysis of the SVM modeling results for the Khoni cultivar, presented in Table 3, reveals that the highest prediction accuracy, with an RPD of 2.7, was attained using the Linear kernel and Gaussian Filter preprocessing. This value places the model in the Excellent category (RPD > 2.5) for accurate quantitative prediction, contrasting sharply with the Linear kernel’s complete failure in the Khormaei cultivar. The non-linear kernels (RBF and Polynomial) also exhibited moderate performance for the Khoni cultivar, with maximum RPD values of 1.58 (with Normalization) and 1.37 (with SNV), respectively. These models are suitable only for qualitative rather than quantitative assessment.

This stark performance difference (the superiority of the Polynomial kernel in Khormaei versus the Linear kernel in Khoni) highlights the disparity in the nature of the spectral-chemical relationships between the two cultivars. It underscores the necessity of employing flexible modeling approaches to individually identify and utilize the optimal combination of model and kernel for each cultivar.

#### 3.1.2. DT Model Performance

The DT model, a non-parametric and non-linear machine learning algorithm, was evaluated for predicting PPO enzyme activity. Complete results are provided in Appendix A, with a summary presented in Table 4.

The DT model results for the Khormaei cultivar demonstrate the best overall performance in this study. The optimal outcome was achieved with MaxSplits = 10 and simple preprocessing methods, including No Preprocessing, Normalization, and Noise Filters (Moving Average, Gaussian Filter, and Median Filter). These models attained an R^2^ of 0.96 and an RPD (Ratio of Performance Deviation) of 4.93 in the test set. This RPD value signifies an excellent model (RPD > 2.5), perfectly suitable for accurate quantitative prediction, and is remarkably superior to the best result achieved by the SVM model for the same cultivar. Increasing the MaxSplits parameter from 5 to 10 resulted in a significant improvement in the test RPD (from 4.73 to 4.93), indicating the necessity for greater complexity to model the spectral details. However, further increases (up to 100) showed no additional effect, suggesting that the optimal complexity had been reached. Unlike the SVM model, the DT model exhibited high stability and accuracy when combined with simple preprocessing techniques. Nevertheless, the application of SNV (Standard Normal Variate) and MSC (Multiplicative Scatter Correction) led to a severe reduction in accuracy, with the RPD Test approaching 1.0, which reconfirms the unsuitability of these specific preprocessing methods for the PPO data.

The performance of the DT model for the Khoni cultivar is also exceptionally strong, on par with the Khormaei cultivar. The highest prediction accuracy, with an R^2^ of 0.96 and an RPD of 5.41, was achieved with MaxSplits = 10 and simple preprocessing (No Preprocessing, Normalization, and Noise Filters) (Table 4). This outcome represents an excellent model, slightly surpassing the best model for the Khormaei cultivar. Increasing MaxSplits from 5 (RPD = 4.21) to 10 (RPD = 5.41) resulted in a substantial leap in accuracy, underscoring the importance of optimizing the complexity parameter for modeling PPO enzyme activity.

#### 3.1.3. PLSR Model Performance

To establish a linear baseline for model evaluation, the performance of the PLSR model on the full spectrum was assessed. The complete details of the PLSR analysis for all preprocessing methods are documented in the Appendix A, while the optimal results are presented in Table 5. The PLSR results for the Khormaei cultivar demonstrate a very weak performance. The best result achieved an RPD of only 0.82 (with 5 components), falling into the ‘Very Poor’ category (RPD < 1.0) and indicating a lack of quantitative predictive power. In stark contrast, the PLSR model for the Khoni cultivar showed a highly competitive performance. The best result, obtained using Normalization preprocessing and 8 components, yielded an R^2^ of 0.86 and an RPD of 2.81. This places the model in the ‘Excellent’ category (RPD > 2.5) for accurate quantitative prediction. This result is the strongest linear model performance observed and is only surpassed by the DT model’s non-linear performance for the same cultivar. The drastic difference in PLSR performance between the two cultivars highlights the varying complexity and nature of the spectral-chemical relationships specific to Khormaei and Khoni.

#### 3.1.4. Overall Discussion and Comparison of Results

The findings of this research unequivocally demonstrate that non-linear models offer robust performance for the non-destructive assessment of PPO enzyme activity via Vis/NIR spectroscopy. The DT algorithm achieved the best absolute performance in this study for the PPO enzyme, yielding RPD values of 4.93 and 5.41 for the Khormaei and Khoni cultivars, respectively. These results fall into the excellent model category, confirming the feasibility of this technology for accurate quantitative prediction of PPO activity.

The inclusion of the PLSR linear model as a baseline provides critical context for these results. As summarized in Table 6, the three modeling approaches exhibited performance ranging from Highly Significant to Not Significant. The complete failure of PLSR for the Khormaei cultivar (RPD = 0.82) strongly indicates that the relationship between the spectrum and PPO activity in this plum is highly non-linear and cannot be adequately modeled by conventional chemometric methods. This conclusion is robustly supported by the statistical analysis, where the PLSR model for Khormaei was the only optimized approach categorized as Not Significant (*p* = 0.081) at the α = 0.05 level. Conversely, while the PLSR model achieved an excellent RPD of 2.81 for the Khoni cultivar, its performance was still significantly inferior to the DT model (RPD = 5.41), demonstrating the clear advantage of non-linear models even when a strong linear component exists. Crucially, the DT models for both cultivars achieved a Highly Significant predictive relationship (*p* < 0.0001), statistically validating their reliability and superior predictive power compared to all other tested models.

The SVM model, despite employing non-linear kernels, exhibited heterogeneous accuracy. Although it reached an excellent level for the Khoni cultivar (RPD = 2.7 with the Linear kernel), its overall performance was weaker compared to the DT model. This suggests that the manner in which tree-based algorithms manage the feature space was more effective for the PPO spectral data in this specific instance.

The consistent observation that preprocessing techniques such as SNV and MSC severely degraded model performance is a crucial finding that warrants discussion. While these methods are highly effective in removing physical light scattering variations caused by sample texture and particle size, they seem to have simultaneously removed or significantly attenuated subtle chemical information highly correlated with PPO enzyme activity. SNV and MSC work by normalizing or centering the entire spectrum to a reference. Our hypothesis is that the unique chemical variance associated with PPO activity was interpreted as simple scattering noise and erroneously removed by these methods, leading to a loss of key predictive features. This finding suggests that for PPO activity prediction in plums, simple Normalization or Noise Filtering is sufficient, as they preserve the crucial baseline shifts and absolute intensity differences that correlate with enzyme concentration.

Furthermore, the models developed for the Khoni cultivar generally demonstrated higher accuracy (particularly with DT) than those for the Khormaei cultivar, which is likely attributable to biochemical and physical differences influencing the spectral signal. These findings are consistent with similar studies in the field of non-destructive monitoring of enzymatic activity in agricultural products and underscore the critical importance of selecting an appropriate non-linear model and meticulously optimizing pre-processing specifically for each quality trait and individual cultivar.

### 3.2. Effective Wavelength Selection

This section of the study focuses on the dimensionality reduction in spectral data using metaheuristic algorithms. The objective is to identify the minimum number of key wavelengths that retain the maximum information required for predicting PPO enzyme activity, while simultaneously reducing computational time and model complexity. The results of the effective wavelength selection by the different algorithms are presented in Figure 3 and Figure 4, showing the performance of the SVM model separately for the Khormaei and Khoni cultivars.

#### 3.2.1. Evaluation Based on Execution Speed

One crucial criterion for evaluating optimization algorithms is their speed in finding an optimal solution. In this study, the execution time for a single run of each algorithm is presented in Table 7. Based on this data, the Particle Swarm Optimization (PSO) algorithm was identified as the fastest among all methods investigated, with approximate execution times of 2.73 s for the Khormaei cultivar and 2.95 s for the Khoni cultivar. This high speed represents a significant advantage in practical applications that require rapid and real-time sample analysis. This finding demonstrates that PSO can effectively locate the optimal wavelengths for data dimensionality reduction within a very short timeframe. While speed is important, the ultimate goal in Feature Selection projects is to achieve the best model performance with the minimum number of features; thus, accuracy and model simplicity retain a higher priority.

#### 3.2.2. Evaluation Based on Prediction Accuracy

The performance of the algorithms in terms of prediction accuracy (evaluated by the lowest mean RMSE and the highest mean Correlation) for the Khormaei and Khoni cultivars is presented in Table 7. For the Khormaei cultivar, the LA algorithm demonstrated the best performance individually with the lowest mean RMSE (≈0.00127), while the GA achieved the highest mean Correlation (≈0.752). However, considering the high number of selected wavelengths (approximately 15 wavelengths) and the requirement for a trade-off between speed and accuracy, the LCA, which selected only 10 wavelengths, is ranked suitably with an RMSE of 0.0032, having eliminated fewer features.

For the Khoni cultivar, the LA algorithm distinctly succeeded in achieving the lowest mean RMSE (≈0.00103) and the highest correlation (≈0.626). These values indicate the superior capability of the LA algorithm to identify wavelengths with the strongest correlation to enzymatic activity in this specific cultivar.

#### 3.2.3. Final Conclusion and Wavelength Selection

Despite the superior execution speed of the Particle Swarm Optimization (PSO) algorithm (less than 3 s) for both cultivars, and its competitive performance in model accuracy, the LA and LCA algorithms yielded excellent results in terms of accuracy metrics (RMSE and Correlation). Specifically, the LCA algorithm, by selecting only 10 wavelengths (the minimum number of selected wavelengths for the Khormaei cultivar), provided satisfactory performance while achieving the maximum dimensionality reduction in the data. Furthermore, for the Khoni cultivar, the LA algorithm demonstrated the best accuracy performance (with the lowest RMSE and highest correlation). Therefore, to achieve the optimal model with the minimum number of wavelengths, a combined approach of SVM-LCA was selected for the Khormaei cultivar, and SVM-LA was chosen for the Khoni cultivar. This selection highlights the structural differences between the two cultivars and the necessity of employing distinct optimization algorithms for the accurate modeling of each. The final selected wavelengths by the SVM-LCA (for Khormaei cultivar) and SVM-LA (for Khoni cultivar) algorithms are presented in Table 8, which will serve as the basis for developing a dedicated and simplified device for the non-destructive monitoring of PPO enzyme activity.

#### 3.2.4. Convergent Analysis of Meta-Heuristic Algorithms

To ensure the robustness of the feature selection process, all meta-heuristic algorithms were run for 30 iterations. The convergence trajectories are presented in Figure 3 and Figure 4. These plots illustrate the change in the fitness value (RMSE_Validation) as the number of iterations increases for different algorithms (PSO, LA, LCA, etc.). As depicted, the algorithms exhibit a rapid initial decrease in RMSE_Validation within the first 10 to 15 iterations. This rapid convergence phase, which might lead to the assumption of a low number of iterations, confirms the high efficiency of the algorithms in quickly isolating the most informative spectral wavelengths. Following this initial steep drop, the curves enter a plateau phase, showing minimal further improvement throughout the remaining iterations. This stable convergence pattern confirms that running the algorithms for 30 iterations was sufficient to achieve a globally optimal or near-optimal solution, ensuring that the final selected wavelength subsets (Table 8) are statistically robust and not merely local optima. This successful convergence is a critical factor for the improved predictive power observed in the DT and SVM models.

#### 3.2.5. Spectral Assignment and Chemical Interpretation of Effective Wavelengths

Analysis of the selected effective wavelengths (Table 8) provides critical insights into the chemical–optical mechanisms exploited by the model. As shown in Table 8, the effective wavelengths for both cultivars are concentrated almost entirely in the Visible (Vis) region of the spectrum (approximately 510–670 nm). Unlike the Near-Infrared (NIR) region, which is dominated by vibrational overtones of O–H and C–H bonds, the Vis region primarily reflects electronic transitions and the absorption characteristics of chromophoric compounds [25].

The strong predictive power of these Vis wavelengths confirms a direct relationship between PPO activity, substrate oxidation, and the resulting color changes associated with enzymatic browning [3]. By selecting wavelengths that correspond to visually detectable pigment transformations, the models effectively captured the spectral signatures of both the early and late stages of the browning reaction. Specifically, the selected spectral regions correspond to:A wavelength of 510–560 nm (Highly Selected Region in Khoni Cultivar)

This wavelength range falls within the green–yellow region of the spectrum and is highly sensitive to the formation of early-stage phenolic oxidation products. Quinone and semiquinone intermediates generated by PPO exhibit characteristic absorption features extending into this region, with some species showing transitions near ~520 nm [26]. Thus, the strong model response within 510–560 nm likely reflects increasing concentrations of these early browning intermediates, rather than carotenoid absorption, which peaks at lower wavelengths (~450 nm) [27].

2.A wavelength of 600–670 nm (Highly Selected Region in Khormaei Cultivar)

This region corresponds to the red-absorption zone, where late-stage brown/black pigments such as melanin-like polymers produce strong broadband absorption. These high-molecular-weight pigments exhibit a monotonic, wide-spectrum absorption profile across the Vis range, with pronounced darkening effects in the red region [28]. Therefore, the 600–670 nm range reliably reflects the advanced stages of enzymatic browning, where quinones polymerize into melanin-like chromophores.

Together, these findings underscore the dominant role of the Visible spectrum in PPO-related browning assessment. The selected wavelengths highlight both the early oxidative transitions (510–560 nm) and late pigment-formation stages (600–670 nm) of the browning pathway, demonstrating that the models successfully isolated the most chemically and optically responsive regions linked to enzymatic browning dynamics.

### 3.3. Discussion and Comparative Analysis of SVM−R and DT Model Performance Based on Full Spectrum and Selected Wavelengths

The performance evaluation of the SVM−R and DT models was conducted across two scenarios—the full spectrum and selected wavelengths—to determine the impact of the dimensionality reduction strategy on the prediction accuracy of PPO enzyme activity.

#### 3.3.1. Investigation of SVM−R Model Performance

##### Performance with Selected Wavelengths and Comparison with Full Spectrum

The SVM−R model demonstrated stable and improved performance following dimensionality reduction using meta-heuristic algorithms. For the Khormaei cultivar, the optimal result was achieved with the Polynomial kernel and Normalization preprocessing, yielding a RPD of 2.65 (Table 9 and Appendix A). This value indicates a marginal improvement compared to the best full-spectrum RPD (RPD = 2.40 with the Polynomial kernel). For the Khoni cultivar, the best SVM−R result was obtained using the Linear kernel and Normalization preprocessing, achieving an RPD of 2.98 (Appendix A). This also represents a slight improvement over the best full-spectrum result (RPD = 2.70 with the Linear kernel).

##### Effect of Dimensionality Reduction

This comparison clearly demonstrates that severe reduction in input dimensionality not only maintained the accuracy of the SVM−R models but, in fact, led to an enhancement in predictive power for both cultivars. This outcome confirms the success of the feature selection algorithm in eliminating irrelevant noise from the full spectrum, consequently enabling the Support Vector Machine model to focus on key spectral information and deliver optimal performance. Notably, following dimensionality reduction, the superior performance of the Linear kernel persisted in the Khoni cultivar, and its performance gap with non-linear kernels (RBF, Polynomial) widened. This suggests a relative linearization of the spectral–chemical relationships after the exclusion of noisy wavelengths.

#### 3.3.2. Evaluation of DT Model Performance

##### Performance with Selected Wavelengths and Comparison with Full Spectrum

The DT model registered the best overall results in this study, leveraging its non-linear capabilities. For the Khormaei cultivar, the optimal DT performance was achieved with a Max Splits = 10 and without the need for preprocessing, yielding an RPD = 3.32 (excellent model) (Table 10 and Appendix A). This outcome represents a marginal decrease compared to the best Full Spectrum RPD (RPD = 4.93). For the Khoni cultivar, the DT model attained the absolute best result across the entire study with an RPD = 5.69 (superb model) (Table 11 and Appendix A). This value indicates a slight improvement over the best Full Spectrum RPD (RPD = 5.41).

**Table 10 foods-14-04297-t010:** Optimal Modeling Results of the DT Method for Khormaei and Khoni Cultivars Following Effective Wavelength Selection.

Variety	Preprocessing	Max Splits	Training			Validation			Test		
			R^2^	RMSE	RPD	R^2^	RMSE	RPD	R^2^	RMSE	RPD
Khormaei	No Preprocessing	5	0.99	0.00026	8.54	0.99	0.00022	9.1	0.9	0.00056	3.22
		10	0.99	0.00024	9.4	0.99	0.00021	9.58	0.9	0.00054	3.32
Khoni	No Preprocessing	5	0.97	0.00029	5.5	0.94	0.00038	4.3	0.94	0.00038	4.38
		10	0.98	0.00023	7	0.97	0.00026	6.37	0.97	0.0003	5.69

**Table 11 foods-14-04297-t011:** Optimal Modeling Results of the PLSR Method for Khormaei and Khoni Cultivars Following Effective Wavelength Selection.

Kernel	Preprocessing	Component	Training			Validation			Test		
			R^2^	RMSE	RPD	R^2^	RMSE	RPD	R^2^	RMSE	RPD
Khormaei	No Preprocessing	9	0.95	0.000449	4.65	0.79	0.00092	2.23	0.44	307	0.96
	Median Filter	9	0.97	0.000345	6.06	0.75	0.000992	2.07	0.42	305	0.97
Khoni	No Preprocessing	8	0.63	0.000953	1.66	0.79	0.000883	2.27	0.38	0.001277	1.05
	Median Filter	8	0.72	0.000826	1.92	0.35	0.00157	1.28	0.44	0.00124	1.08

##### Impact of Dimensionality Reduction

The superiority of the DT model in predicting PPO activity over the SVM model was consistently observed with a significant margin, establishing DT as the most robust algorithm.

This stark difference stems from DT’s intrinsic properties: As an inherently non-linear and non-parametric model, DT better models the complex, non-uniform, and threshold-based relationships within the spectral data by establishing local decision rules, contrasting with SVM, which primarily seeks one global optimal hyperplane. Furthermore, DT demonstrated greater flexibility in managing high collinearity within the spectral data and exhibited higher stability against residual noise, achieving its best performance often with simple or no pre-processing. This high stability confirms the superior informational power of the selected key wavelengths.

The dimensional reduction in the input data by over 90% constituted a favorable operational compromise. While it slightly decreased the RPD in the Khormaei cultivar (though remaining in the excellent category), it simultaneously significantly increased the efficiency in the Khoni cultivar by eliminating interfering noise, thereby proving the exceptional efficacy of the feature selection process. These results are fully consistent with the findings of other relevant studies concerning the use of metaheuristic algorithms for optimal feature selection. For instance, in a study conducted by Taghinezhad et al. (2025) on rice, which utilized NIR spectroscopy for the non-invasive identification of starch gelatinization, emphasized the importance of dimensionality reduction for enhancing model efficiency [29].

#### 3.3.3. Examination of PLSR Model Performance

In contrast to the non-linear models, the PLSR linear model’s performance significantly degraded following dimensionality reduction (Table 11 and Appendix A). For the Khormaei cultivar, the optimal PLSR result achieved an RPD of only 0.97 (Very Poor), and for the Khoni cultivar, the RPD was 1.08 (Weak). This outcome demonstrates that the small subset of selected wavelengths, while highly informative for the non-linear DT and SVM models, did not contain sufficient comprehensive covariance structure necessary for the linear regression foundation of PLSR. Crucially, statistical analysis confirmed that the PLSR results for both Khormaei (*p* = 0.612) and Khoni (*p* = 0.64) were Not Significant, indicating that the correlation established by PLSR after feature selection was no better than a random guess.

#### 3.3.4. Overall Comparison and Implications of Dimensionality Reduction

The overall best performance achieved by each model following feature selection is summarized in Table 12. These comparisons clearly indicate that metaheuristic algorithms such as Particle Swarm Optimization (PSO) are powerful tools for the optimization of chemometric models. The DT model stands out as the superior algorithm, achieving RPD values up to 5.69, which represents a highly reliable and accurate quantitative prediction system. By overcoming the challenges associated with voluminous and highly correlated spectral data, these algorithms pave the way for a broader and more practical application of non-destructive spectroscopy in the food industry. Furthermore, the robust statistical significance (*p* < 0.0001) achieved by the best DT and SVM models (Table 12) strongly contrasts with the statistical insignificance of all PLSR models post-feature selection, unequivocally supporting the adoption of non-linear machine learning approaches for complex enzymatic activity assessment.

## 4. Discussion

The findings of this research, which establish the feasibility of non-destructive detection of PPO enzyme activity in plum cultivars using a combination of Vis/NIR spectroscopy and machine learning algorithms, hold significant importance for the development of rapid quality control systems in the food industry. The obtained results, particularly the excellent performance of the DT model with RPD values exceeding 4.9 in the full-spectrum range for both plum cultivars, not only confirms the high reliability of this methodology but also positions it among the best-reported performances in this domain.

In a similar study utilizing Vis/NIR spectroscopy and a metaheuristic approach (DT-FOA) to assess PPO and POD activity in ‘Red Delicious’ apples, the ANN model yielded the best result for PPO with an RPD = 4.87. This RPD value is close to the magnitudes obtained in the present study and reinforces the superiority of non-linear models such as DT and ANN over linear regressions (e.g., PLSR and MLR) for predicting chemical and biochemical properties of various fruits [5].

Furthermore, in a study evaluating PPO activity in bell pepper, the combination of a metaheuristic algorithm with the ANN model demonstrated superior performance compared to linear regression methods such as PLSR and MLR [9]. This finding aligns with the results of the current study, indicating that non-linear regression models perform more accurately for processing complex spectral data, thereby enabling the modeling of intricate relationships between spectral data and enzymatic activity.

A study on lychee fruit peel using Hyperspectral Imaging (HSI) for PPO prediction concluded that the combination of spectral information and image features (such as color parameters), alongside a Fuzzy Neural Network (FNN), yielded the highest accuracy (R^2^ up to 0.91) [8]. Although the current study focused solely on Vis/NIR spectral data, its results indicate that the Visible (Vis) spectrum, which directly correlates with color parameters, alone contains critical and sufficient information for the non-destructive assessment of PPO in plum.

Even in cases where spectroscopy was not directly employed, such as the assessment of PPO in banana peel using Digital Image Processing and Genetic Programming (GP), a high correlation (R = 0.98) was observed between color parameters and PPO activity [7]. This confirms that optical measurements in the Visible (Vis) range, whether via spectroscopy or imaging, are significantly related to the enzymatic browning process.

One of the key findings of this research was achieving the highest accuracy using a subset of key wavelengths. This aligns with the general trend in chemometrics and machine learning, as other studies also emphasize that the selection of Effective Wavelengths (EWs) using optimization algorithms (such as SVM−PSO in bell pepper) contributes to the improved performance of regression models.

The remarkably high performance of the Decision Tree (DT) model observed in this study warrants further discussion. Although non-linear models are generally expected to outperform linear regression methods when modeling complex biochemical phenomena, the consistently high RPD values of DT, particularly for the Khoni cultivar, highlight the unique capability of tree-based algorithms in capturing non-linear relationships in spectral data. The use of metaheuristic algorithms for selecting key wavelengths substantially reduced noise and emphasized the most informative spectral bands. DT models are particularly sensitive to the quality of input features, and this optimized selection likely contributed to the model’s robustness and high predictive power.

DT inherently handles complex interactions between predictor variables, which is crucial when modeling PPO activity that depends on subtle combinations of spectral features influenced by enzymatic browning and phenolic content. Similar trends have been reported in previous studies, where tree-based or non-linear models outperformed linear approaches. For instance, a recent study comparing decision tree-based algorithms (Decision Tree, Random Forest, and XGBoost) for food discrimination using NIR and Raman spectroscopy reported very high accuracies of up to 99% [30].

The difference in DT performance between the Khoni and Khormaei cultivars reflects the model’s ability to capture cultivar-specific spectral signatures. Previous literature suggests that the performance of chemometric models, including non-linear ones, often depends on sample matrix variability and cultivar or species differences [31].

Although methods such as Support Vector Machine Regression (SVM-R), Artificial Neural Networks (ANN), or PLSR have been widely applied to Vis/NIR spectral data, tree-based algorithms offer advantages in interpretability and computational efficiency, especially when combined with effective wavelength selection. For example, in a study evaluating black tea quality via hyperspectral imaging and multiple decision-tree models, a Fine Tree model achieved a correct classification rate of 93.13% [32]. This demonstrates that DT can serve as a reliable alternative for rapid and accurate non-destructive assessment in industrial applications.

In summary, the exceptional performance of DT in this study not only confirms its suitability for non-linear regression of spectral data but also emphasizes the importance of proper feature selection and cultivar-specific modeling. Future investigations could explore ensemble tree methods, such as Random Forest or Gradient Boosting, to further enhance prediction robustness and generalizability across multiple cultivars.

### 4.1. Numerical Comparison with Existing Literature

To contextualize the strong performance achieved in this study, the results were numerically benchmarked against similar reports focused on enzymatic activity and quality assessment in fruit using Vis/NIR spectroscopy (Table 13). While the textual comparison in the preceding paragraphs establishes the general superiority of non-linear models, the structured data in Table 13 provides quantitative proof. For example, our best result (RPD = 5.69 for Khoni cultivar using DT) significantly surpasses other quantitative prediction efforts for biochemical parameters in fruits, including the highly successful RPD = 4.87 reported for PPO in apples [5]. Furthermore, the RPD achieved here is notably higher than the typical threshold of 3.0 used for reliable quantitative analysis. While studies using different metrics, such as computer vision [7] (R^2^ = 0.98) or Hyperspectral Imaging [8] (R^2^ = 0.91), also show strong correlations, our high RPD value confirms the superior efficacy of our optimized non-linear DT approach coupled with metaheuristic feature selection in isolating the most relevant chemical information related to PPO activity in plums. This successful benchmarking reinforces the potential of this methodology for industrial quality control.

### 4.2. Impact of Spectral Preprocessing

A crucial methodological finding of this study relates to the influence of spectral preprocessing techniques. As detailed in the results (Table 9, Table 10, Table 11 and Table 12), simple methods such as Normalization and the application of Median Filtering for noise reduction consistently yielded the best model performance. Normalization successfully corrected for global physical baseline shifts and differences in light scattering path length, which is a common variability source in diffuse reflectance measurements.

However, more complex scattering correction methods, specifically Standard Normal Variate (SNV) and Multiplicative Scatter Correction (MSC), resulted in a significant degradation of accuracy across all models and cultivars. The failure of SNV and MSC suggests a critical chemometric phenomenon: these methods may have erroneously removed or suppressed the subtle but crucial chemical variance directly linked to the concentration changes in PPO substrates and products. Since PPO activity is strongly coupled to the formation of light-absorbing pigments (chromophores) in the visible range, the complex mathematical transformations imposed by SNV and MSC likely blurred the essential non-linear relationship between the spectra and the enzymatic activity. This result underscores the need for careful empirical selection of preprocessing methods, favoring simple approaches that preserve the underlying chemical signal.

### 4.3. Influence of Cultivar-Specific Factors on Prediction

A notable finding is the distinct difference in prediction performance observed between the two plum cultivars. The Khoni cultivar yielded the best result (RPD = 5.69), while the Khormaei cultivar achieved a slightly lower, though still excellent, result (RPD = 4.93). This variation can be attributed to inherent cultivar-specific matrix effects and structural differences. Firstly, physical factors such as skin thickness, firmness, and microstructure lead to differences in light scattering properties, affecting the photon penetration depth and signal intensity, which requires the model to learn distinct relationships for each variety. Secondly, the initial biochemical matrix composition differs between the two cultivars. Since the models rely on the indirect correlation between the PPO-induced color change and the Vis spectrum, a difference in the initial phenolic substrate profile or background pigmentation will inevitably affect the predictive model’s sensitivity. This necessitates the use of separate, customized models for each cultivar, rather than a single generalized model, to maintain the high predictive performance required for industrial application.

## 5. Conclusions

The results from modeling PPO enzyme activity in the Khormaei and Khoni plum cultivars, utilizing Vis/NIR spectroscopy and Machine Learning algorithms, clearly substantiate the superiority of the non-linear approach and dimensionality reduction.

The prominence of the Decision Tree (DT) and Support Vector Machine Regression (SVM-R) models demonstrated significantly stronger performance compared to linear methods such as Partial Least Squares Regression (PLSR). This confirms that the relationship between Vis/NIR spectral absorbance and PPO enzyme activity is of an inherently complex and non-linear nature. The poor and statistically insignificant performance of PLSR, both with the full spectrum and selected wavelengths (RPD down to 0.97), strongly supports the necessity of employing non-linear machine learning for this application. The DT model, achieving the highest RPD values in both cultivars (up to 5.41 in the full spectrum), is established as the most robust algorithm for PPO prediction.

The strategy of dimensionality reduction to approximately 15 selected wavelengths proved to be a successful solution. In the SVM-R model, dimensionality reduction led to a marginal improvement in both cultivars (from 2.40 to 2.65 for Khormaei and from 2.70 to 2.98 for Khoni). For the DT model applied to the Khoni cultivar, the accuracy was not only maintained but further enhanced (from RPD = 5.41 to RPD = 5.69). This stability and improvement in accuracy, despite the substantial reduction in input data, confirms that the selected wavelengths possess minimal noise and maximum informational power.

The DT model using selected wavelengths for the Khoni cultivar (RPD = 5.69 and R^2^ = 0.97) is proposed as the best and most efficient option for applications requiring accurate and quantitative quality monitoring. The optimal DT performance in the unprocessed state with selected wavelengths guarantees the simplicity and speed of execution for the final model. The necessity of building cultivar-specific models was also highlighted by the significant performance gap observed between the two cultivars. These findings represent a crucial step toward the transfer of filter spectroscopy technology to industry, enabling the use of cheaper and faster sensors for real-time monitoring of enzymatic activity and non-destructive quality control within the plum supply chain.

## Figures and Tables

**Figure 1 foods-14-04297-f001:**
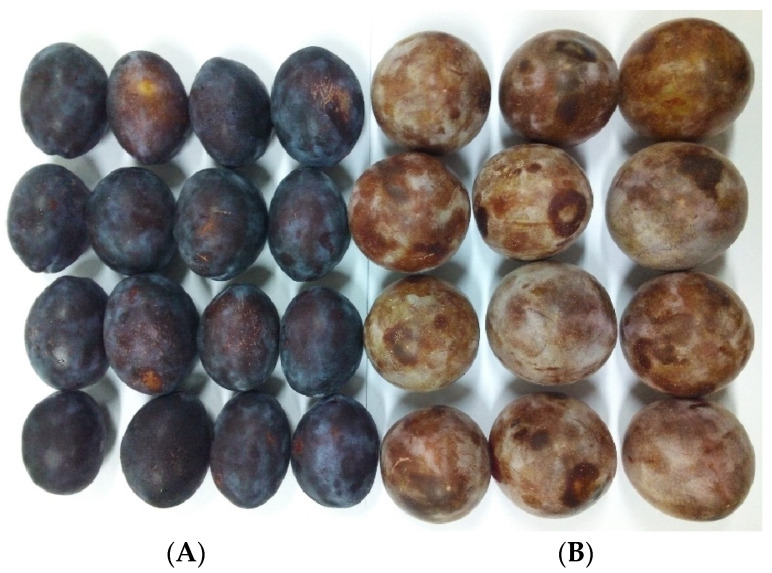
Representative plum cultivars employed: (**A**) Khormaei cultivar and (**B**) Khoni cultivar.

**Figure 2 foods-14-04297-f002:**
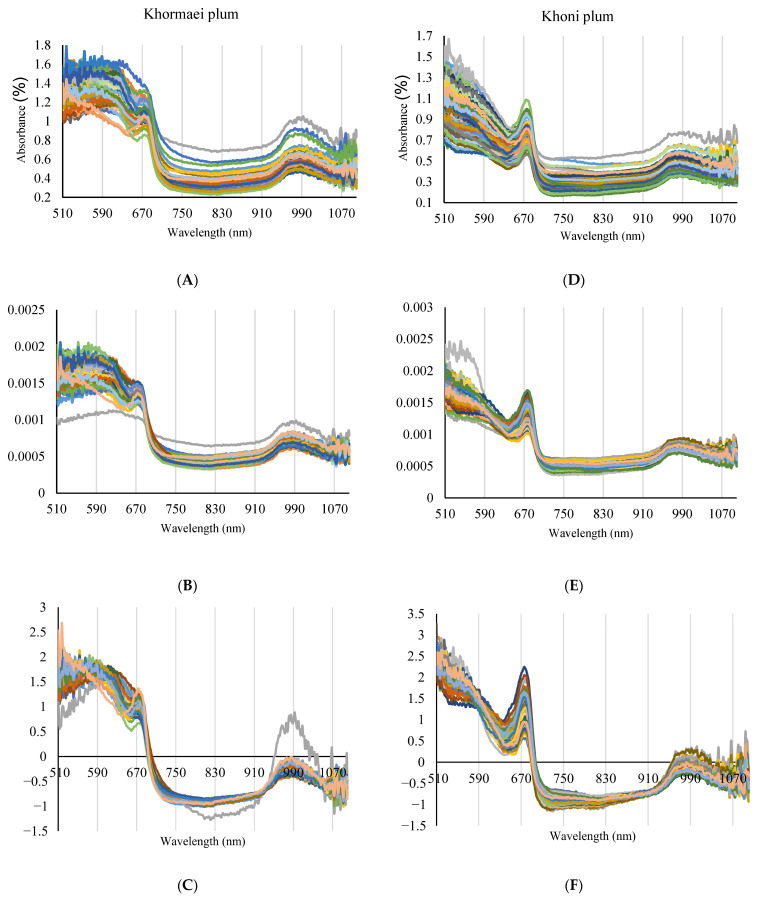
Visual Comparison of Raw (**A**,**D**), Normalized (**B**,**E**), and SNV Preprocessed (**C**,**F**) Vis/NIR Spectra Across Khormaei and Khoni Cultivars.

**Figure 3 foods-14-04297-f003:**
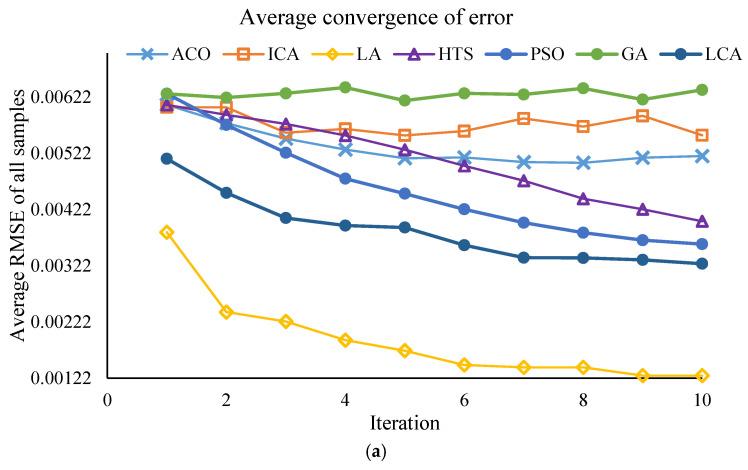
Performance of the SVM model with different algorithms for Khormaei plum variety samples: (**a**) Average RMSE for all samples, (**b**) Average correlation for all samples.

**Figure 4 foods-14-04297-f004:**
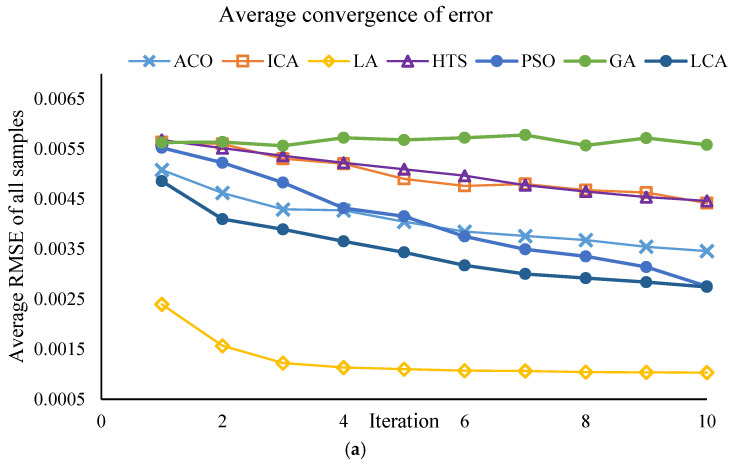
Performance of the SVM model with different algorithms for Khoni plum variety samples: (**a**) Average RMSE for all samples, (**b**) Average correlation for all samples.

**Table 1 foods-14-04297-t001:** Specifications of the Hyperparameters for Each Effective Wavelength Selection Algorithm.

Row	Algorithms	Parameters
1	LCA	Teams: 8, Seasons: 10, Match Time: 2
2	GA	Population: 120, Generations: 10, Crossover Rate: 30%, Mutation Rate: 30%
3	PSO	Particles: 40, Episodes: 10, P Bests: 8
4	ACO	Ants: 40, Episodes: 10, Evaporation Rate: 4
5	ICA	Countries: 40, Imperialists: 11
6	LA	Generations: 10, Tries: 40
7	HTS	Molecules: 15, Iterations: 10, CDF: 2, COF: 10, RDF: 2

**Table 2 foods-14-04297-t002:** Descriptive Statistics for PPO Activity (absorbance/min·g) Across Various Plum Cultivars.

Variety	Mean	Standard Deviation	Maximum	Minimum
Khoni	0.003165	0.001614	0.005849	0.00012
Khormaei	0.003256	0.002062	0.006374	0.000145

**Table 3 foods-14-04297-t003:** Optimal Modeling Results of the SVM-R Method in the Khormaei and Khoni Cultivars.

Variety	Kernel	Preprocessing	Training			Validation			Test		
			R^2^	RMSE	RPD	R^2^	RMSE	RPD	R^2^	RMSE	RPD
Khormaei	RBF	No Preprocessing	0.88	0.0007	2.92	0.57	0.00123	1.58	0.57	0.00149	1.57
		Median Filter	0.91	0.00062	3.32	0.62	0.00115	1.69	0.63	0.00138	1.7
	Polynomial	No Preprocessing	0	0.00203	1.01	0.08	0.00194	1	0	0.00227	1.03
		Normalization	0.98	0.00027	7.64	0.74	0.00095	2.03	0.81	0.00098	2.4
Khoni	Linear	No Preprocessing	0.99	0.00017	9.29	0.97	0.00028	6.11	0.85	0.00065	2.68
		Moving Average	0.99	0.00017	9.09	0.97	0.00028	6.1	0.85	0.00065	2.67
		Gaussian Filter	0.99	0.00017	9.06	0.97	0.00028	6.13	0.85	0.00064	2.7
		Median Filter	0.99	0.00017	9.32	0.97	0.00029	5.93	0.85	0.00065	2.68

**Table 4 foods-14-04297-t004:** Optimal Modeling Results of the DT Method for the Khormaei and Khoni Plum Cultivars.

Variety	Preprocessing	Max Splits	Training			Validation			Test		
			R^2^	RMSE	RPD	R^2^	RMSE	RPD	R^2^	RMSE	RPD
Khormaei	No Preprocessing	5	0.98	0.00026	7.74	0.99	0.00023	9.77	0.95	0.00044	4.73
		10	0.99	0.00023	8.61	0.99	0.00019	11.57	0.96	0.00042	4.93
Khoni	No Preprocessing	5	0.97	0.00028	6.36	0.94	0.00031	4.31	0.94	0.00034	4.21
		10	0.99	0.00018	9.77	0.97	0.00022	6.08	0.96	0.00026	5.41

**Table 5 foods-14-04297-t005:** Optimal Modeling Results of the PLSR Method in the Khormaei and Khoni Cultivars.

Kernel	Preprocessing	Component	Training			Validation			Test		
			R^2^	RMSE	RPD	R^2^	RMSE	RPD	R^2^	RMSE	RPD
Khormaei	No Preprocessing	5	0.87	0.000729	2.75	0.88	0.000658	3	0.58	0.002874	0.82
	Mean Centering	5	0.87	0.000729	2.75	0.88	0.000658	3	0.58	0.002874	0.82
Khoni	No Preprocessing	8	0.98	0.000207	7.55	0.97	0.000284	6.33	0.82	0.000713	2.42
	Normalization	8	0.98	0.000225	6.92	0.96	0.000331	5.44	0.86	0.000615	2.81

**Table 6 foods-14-04297-t006:** Optimal Performance Comparison of Three Modeling Aproaches.

Variety	Model	Optimal Preprocessing/Kernel	Classification	*p*-Value	Significance (α = 0.05)
Khormaei	DT	Max Splits 10	Excellent	<0.0001	Highly Significant
	SVM	Polynomial, Normalization	Very Good	0.003	Significant
	PLSR	No Preprocessing/Mean Centering	Very Poor	0.081	Not Significant
Khoni	DT	Max Splits 10	Excellent	<0.0001	Highly Significant
	PLSR	Normalization	Excellent	0.009	Significant
	SVM	Linear, Gaussian Filter	Excellent	0.001	Highly Significant

**Table 7 foods-14-04297-t007:** Comparison of Meta-heuristic Algorithm Performance in Wavelength Selection for PPO Activity Prediction.

Algorithm	No. of Selected Features	Khormaei			Khoni		
		Average RMSE	Average Correlation	Run Time (s)	Average RMSE	Average Correlation	Run Time (s)
PSO	15	0.003605	0.29469	2.73	0.002756	0.61479	2.95
LCA	10	0.003257	0.33528	88.45	0.002744	0.61136	109.33
GA	14	0.006348	0.75224	2.99	0.005581	0.53946	3.14
ICA	15	0.00554	0.66917	2.84	0.00442	0.58546	2.83
LA	13	0.001268	0.013096	62.37	0.001033	0.62579	59.13
HTS	15	0.004012	0.50522	24.01	0.004462	0.58635	25.27
ACO	15	0.00517	0.6684	56.58	0.003461	0.58979	56.26

**Table 8 foods-14-04297-t008:** Selected wavelengths of the SVM-LA and SVM-LCA algorithm.

Variety	Algorithm	Selected EWs (nm)
Khormaei	SVM-LCA	604, 633, 648, 629, 549.5, 586.5, 625, 631, 667, 641.5
Khoni	SVM-LA	544.5, 524, 543.5, 540, 518.5, 512, 519, 560.5, 523.5, 544, 554.5, 514.5, 525, 534.5, 526

**Table 9 foods-14-04297-t009:** Optimal Modeling Results of the SVM-R Method for Khormaei and Khoni Cultivars Following Effective Wavelength Selection.

Variety	Kernel	Preprocessing	Training			Validation			Test		
			R^2^	RMSE	RPD	R^2^	RMSE	RPD	R^2^	RMSE	RPD
Khormaei	RBF	No Preprocessing	0.95	0.00044	4.74	0.5	0.00144	1.46	0.71	0.00085	1.93
		Mean Centering	0.86	0.00078	2.71	0.36	0.00164	1.29	0.56	0.00106	1.56
	Polynomial	No Preprocessing	0	0.00208	1.01	0.02	0.00206	1.02	0.64	0.00204	0.81
		Normalization	0.29	0.00523	0.4	0.45	0.00246	0.86	0.85	0.00062	2.65
Khoni	Linear	No Preprocessing	0.98	0.0002	7.99	0.93	0.00042	3.86	0.86	0.00059	2.78
		Normalization	0.98	0.0002	7.89	0.92	0.00045	3.56	0.88	0.00055	2.98
		Median Filter	0.98	0.0002	8.14	0.93	0.00041	3.92	0.87	0.00058	2.84
		Mean Centering	0.98	0.0002	8.1	0.93	0.00041	3.94	0.86	0.00059	2.8

**Table 12 foods-14-04297-t012:** Optimal Performance Comparison of Models after Feature Selection.

Variety	Model	Optimal Preprocessing/Kernel	Classification	*p*-Value	Significance (α = 0.05)
Khormaei	DT	Max Splits 10	Excellent (RPD = 3.32)	<0.0001	Highly Significant
	SVM	Polynomial, Normalization	Excellent (RPD = 2.65)	0.001	Significant
	PLSR	Median Filter	Very Poor (RPD = 0.97)	0.612	Not Significant
Khoni	DT	Max Splits 10	Excellent (RPD = 5.69)	<0.0001	Highly Significant
	SVM	Linear, Normalization	Excellent (RPD = 2.98)	0.0003	Highly Significant
	PLSR	Median Filter	Weak (RPD = 1.08)	0.64	Not Significant

**Table 13 foods-14-04297-t013:** Numerical Comparison of the Best Predictive Models for Enzymatic Activity and Quality Assessment in Fruits Using Vis/NIR Spectroscopy.

Reference	Product	Target Feature	Spectroscopy Technique	Model	Accuracy Metric (e.g., RPD, R^2^)
[5]	Apple	PPO	Vis/NIR	ANN	R^2^ = 0.96, RPD = 4.87
[8]	Lychee	PPO	HSI	FNN	R^2^ = 0.91
[9]	Bell Pepper	PPO	Vis/NIR	ANN	R^2^ = 0.90, RPD = 3.86
[7]	Banana	PPO	Computer Vision System	GP	R^2^ = 0.98
This Study	Khoni Plum	PPO Activity	Vis/NIR + Metaheuristic	DT	R^2^ = 0.97, RPD = 5.69

## Data Availability

The original contributions presented in this study are included in the article/Appendix A. Further inquiries can be directed to the corresponding authors.

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
