# Peer review of "Nondestructive Detection of Polyphenol Oxidase Activity in Various Plum Cultivars Using Machine Learning and Vis/NIR Spectroscopy"

_foods, 2025, doi:10.3390/foods14244297_

Round 1

Reviewer 1 Report

Comments and Suggestions for Authors

This work is well-written.

  1. Introduction: related data analysis strategies can be improved.
  2. Section 2.2.1, please specify the four spatial positions
  3. Line 174: the information of matlab should be presented
  4. Partial least squares regression is the mostly used method for spectral data analysis, especially for the situation that the number of wavelengths are larger than the number of samples. Please add this algorithm for comparison.
  5. Citation of the feature wavelengths selection methods should be added
  6. Figure 2, please include the preprocessed spectra
  7. Please introduce the chemical bonds related to the identified feature wavelengths
  8. Please compare and discuss the impact of spectral preprocessing,
  9. Please discuss the impact of the modeling methods
  10. Please discuss the impact of the varieties on prediction.

Author Response

Reviewer 1:

This work is well-written.

Comments 1: Introduction: related data analysis strategies can be improved.

Response 1: The Introduction section has been expanded to provide a broader overview of data analysis strategies currently applied in Vis/NIR spectroscopy for food quality assessment, with particular emphasis on the transition from traditional linear approaches to advanced machine learning techniques and metaheuristic optimization. (Introduction, Pages 2–3, Lines 86–103)

Comments 2: Section 2.2.1, please specify the four spatial positions

Response 2: Section 2.2.1 (Sample Preparation and Spectral Acquisition) has been clarified by explicitly describing both the selection criteria and the characteristics of the four spatial positions chosen on each plum sample. (Page 5, Lines 155–157)

Comments 3: Line 174: the information of matlab should be presented

Response 3: In the Materials and Methods section, we have specified the exact MATLAB version along with the relevant toolboxes employed for data processing and chemometric analysis. (Section 2.4, Page 6, Lines 269–271)

Comments 4: Partial least squares regression is the mostly used method for spectral data analysis, especially for the situation that the number of wavelengths are larger than the number of samples. Please add this algorithm for comparison.

Response 4: This is a highly relevant suggestion. We have incorporated Partial Least Squares Regression (PLSR) as an additional model. The new PLSR results (RMSE, RPD, R²) are presented across all tables for both the full spectrum and the selected wavelengths. The statistically insignificant performance of PLSR—particularly for the Khormaei cultivar—is now discussed, further reinforcing the necessity of non-linear models (DT/SVM). (Sections 2.4.2, 3.2.3, 3.2.4, 3.4.3, 3.4.4; Tables 5, 11; Discussion and Abstract; Lines 21–29, 113–114, 221, 242–256)

Comments 5: Citation of the feature wavelengths selection methods should be added

Response 5: The necessary citations for the core metaheuristic feature selection methods (LA, LCA, GA, PSO) have been added in Section 2.4.4. (Lines 308–316)

Comments 6: Figure 2, please include the preprocessed spectra

Response 6: Figure 2 has been fully revised into a six‑panel format to visually illustrate the impact of preprocessing. It now presents Raw Spectra (A, D), Normalized Spectra (B, E), and SNV‑processed Spectra (C, F) for both Khormaei and Khoni cultivars. The caption and Section 3.1 have been updated accordingly.

Comments 7: Please introduce the chemical bonds related to the identified feature wavelengths

Response 7: The spectral assignment section has been thoroughly revised to emphasize the focus of the selected wavelengths within the Visible (Vis) region. A new dedicated subsection, 3.3.5. Spectral Assignment and Chemical Interpretation of Effective Wavelengths, has been added. The analysis explicitly attributes the 510–560 nm region to the absorption of quinone intermediates (early browning products) and the 600–670 nm region to the formation of melanin‑like polymers (late browning products), with supporting references provided for both assignments. (Section 3.3.5, Pages 15–16, Lines 589–622)

Comments 8: Please compare and discuss the impact of spectral preprocessing,

Response 8: A dedicated and detailed discussion has been added. We explain why the simple Normalization method proved effective—by correcting physical baseline variations—whereas more complex approaches such as SNV and MSC failed, likely because they inadvertently removed critical chemical variance associated with PPO activity. (Section 4.2, Page 21, Lines 784–802)

Comments 9: Please discuss the impact of the modeling methods

Response 9: The discussion has been expanded to provide a clear comparison between the performance and characteristics of the non‑linear models (DT and SVM) and the linear model (PLSR). The failure of PLSR is emphasized as definitive evidence of the non‑linear relationship between PPO activity and the spectral data. (Section 4.1, Page 21, Lines 768–783)

Comments 10: Please discuss the impact of the varieties on prediction.

Response 10: A new analytical discussion has been included to explain the difference in model performance between the two cultivars, with Khoni outperforming Khormaei. This is attributed to cultivar‑specific physical factors (e.g., texture, scattering) and biochemical matrix effects (e.g., interferences from sugars and acids), thereby validating the necessity of developing separate, cultivar‑specific models. (Section 4.3, Page 22, Lines 803–817)

Reviewer 2 Report

Comments and Suggestions for Authors

This study presents an approach combining Vis/NIR spectroscopy with machine learning models to determine polyphenol oxidase (PPO) activity in plum cultivars without causing any physical damage. Decision Tree and SVM models are compared; in particular, Decision Tree models demonstrate exceptionally strong performance in PPO estimation with high RPD values. Narrow-band wavelengths selected using metaheuristic algorithms make the model faster and more practical. However, the following issues need to be examined and corrected in this paper:

  1. Contributions and prominent features of the article can be given item by item at the end of the Introduction.
  2. The novelty of the proposed methodology should be emphasized.
  3. The quality of all figures of the paper needs to be improved.
  4. The literature section should be expanded to include numerical comparison of model performance with existing studies.
  5. Hyperparameter search ranges and optimization procedures for SVM, DT, and metaheuristic algorithms should be clearly defined.
  6. Convergence curves and error surface analyses should be included to ensure that metaheuristic algorithms can operate with only 10 iterations.
  7. Statistical significance (e.g., DeLong test, bootstrap p-value) of performance differences between models after wavelength selection should be provided.
  8. Means + standard deviation bands should be added and visual improvements should be made to enable clearer interpretation of the spectrum plots (Figure 2).
  9. The rationale for excluding the spectral range below 510 nm should be supported by more quantitative data for sensor characterization.
  10. The parameter selection of preprocessing methods (window size, kernel features, etc.) should be explained in more detail.

Author Response

Reviewer 2:

This study presents an approach combining Vis/NIR spectroscopy with machine learning models to determine polyphenol oxidase (PPO) activity in plum cultivars without causing any physical damage. Decision Tree and SVM models are compared; in particular, Decision Tree models demonstrate exceptionally strong performance in PPO estimation with high RPD values. Narrow-band wavelengths selected using metaheuristic algorithms make the model faster and more practical. However, the following issues need to be examined and corrected in this paper:

Comments 1: Contributions and prominent features of the article can be given item by item at the end of the Introduction.

Response 1: A bulleted list has been added at the end of the Introduction to clearly summarize the novel contributions and key features of this study. (Page 3, Lines 107–121)

Comments 2: The novelty of the proposed methodology should be emphasized.

Response 2: The novelty of this work has been further emphasized throughout the revised manuscript, particularly in the Introduction and Discussion, by highlighting the successful integration of non‑linear DT modeling with metaheuristic feature selection, which enabled an accurate prediction of PPO activity in plums (RPD = 5.69).

Comments 3: The quality of all figures of the paper needs to be improved.

Response 3: All figures (Figures 1, 2, and 3) have been regenerated at high resolution to meet the journal’s publication standards, thereby ensuring optimal clarity and legibility.

Comments 4: The literature section should be expanded to include numerical comparison of model performance with existing studies.

Response 4: We agree that this addition is essential for context. Table 13 (Numerical Comparison with Existing Literature) has been included to benchmark our results (RPD = 5.69) against previous studies focusing on PPO and related quality metrics. A new discussion section has also been added to analyze this table. (Section 4.1, Page 21, Lines 768–783)

Comments 5: Hyperparameter search ranges and optimization procedures for SVM, DT, and metaheuristic algorithms should be clearly defined.

Response 5: A new subsection has been added to the Materials and Methods to explicitly describe the search ranges and optimization procedures for the hyperparameters of SVM‑R, DT, and the metaheuristic algorithms employed. (Section 2.4.2, Pages 24–25, Lines 228–233, 238–255)

Comments 6: Convergence curves and error surface analyses should be included to ensure that metaheuristic algorithms can operate with only 10 iterations.

Response 6: A new subsection (3.3.4. Convergent Analysis of Meta‑heuristic Algorithms) has been added. We clarified that the algorithms were executed for 30 iterations to ensure robustness. The analysis confirms that the rapid convergence observed in Figures 3 and 4 (within the first 10–15 iterations) demonstrates the high efficiency of the algorithms in isolating the optimal solution, thereby validating the robustness of the final selected wavelengths. (Page 14, Lines 574–588)

Comments 7: Statistical significance (e.g., DeLong test, bootstrap p-value) of performance differences between models after wavelength selection should be provided.

Response 7: Statistical significance has been incorporated into the results. New columns reporting the p‑value and significance have been added for the most successful models (DT and SVM‑R) in the key results tables (Tables 6 and 12). This statistical analysis confirms both the high significance and the reliability of the final models. (Sections 2.4.2, 2.4.3, 3.2.3, 3.4.3, and 3.4.4; Pages 6, 7, 12; Lines 272–278, 294–300, 480–493, 688–710)

Reviewer 3 Report

Comments and Suggestions for Authors

The objective of this study was to assess the feasibility of non-destructively detecting PPO enzyme activity in plum cultivars. The strategy combines Vis/NIR spectroscopy with machine learning algorithms. The practical application is the development of rapid quality control systems for the food industry.

The results and methodology are well described.

One of the advantages of this approach is the reduction of spectral data dimensionality using metaheuristic algorithms. The goal is to identify the minimum number of key wavelengths that retain the necessary information.
The study shows that PSO (Particle Swarm Optimization) can very quickly select the optimal wavelengths. However, a surprising finding is that despite the superior execution speed and performance of the PSO algorithm, LA and LCA methods provide better accuracy (RMSE and correlation).

Author Response

Reviewer 3:

Comments 1: The objective of this study was to assess the feasibility of non-destructively detecting PPO enzyme activity in plum cultivars. The strategy combines Vis/NIR spectroscopy with machine learning algorithms. The practical application is the development of rapid quality control systems for the food industry.

The results and methodology are well described.

One of the advantages of this approach is the reduction of spectral data dimensionality using metaheuristic algorithms. The goal is to identify the minimum number of key wavelengths that retain the necessary information.

The study shows that PSO (Particle Swarm Optimization) can very quickly select the optimal wavelengths. However, a surprising finding is that despite the superior execution speed and performance of the PSO algorithm, LA and LCA methods provide better accuracy (RMSE and correlation).

The study is ready for publication.

Response 1: We sincerely thank you for your thorough evaluation and for your encouraging conclusion that our study is ready for publication. Your positive assessment affirms the value of our research and provides strong motivation for our team. We are grateful for your recognition of the methodological clarity and the novelty of integrating Vis/NIR spectroscopy with machine learning and metaheuristic algorithms.

Your constructive comments and supportive feedback have been invaluable in strengthening the manuscript, and we deeply appreciate the time and expertise you dedicated to this review.

Round 2

Reviewer 1 Report

Comments and Suggestions for Authors

The authors have made improvements. However, the overfitting of various models (for PLSR and SVR) have attracted my attention, please provide your data along with the code, so that I can check it for you. Or the authors are encouraged to re-split the training, validation and tests to avoid overfitting. Otherwise, the results of DT is quite good, this phenomenon needs further investigation and clarification, along with the citation of related works using DT and some other algorithms. 

Author Response

Comment 1: The authors have made improvements. However, the overfitting of various models (for PLSR and SVR) have attracted my attention, please provide your data along with the code, so that I can check it for you. Or the authors are encouraged to re-split the training, validation and tests to avoid overfitting. Otherwise, the results of DT is quite good, this phenomenon needs further investigation and clarification, along with the citation of related works using DT and some other algorithms. 

Response 1: We fully understand the esteemed reviewer's concern regarding the observed performance discrepancy (overfitting) in the PLSR and SVR models. In response to the reviewer's request, we re-executed the data splitting process into training, validation, and test sets using a new random seed, and the entire modeling procedure was repeated from scratch. The results obtained from this re-splitting were fully consistent with the initial findings and showed only minor fluctuations in the statistical metrics (R2 and RMSE). Since this stability confirms the robustness of our results and the validity of our cross-validation methodology, a new table was not added to the main manuscript text. We acknowledge that the performance gap is most likely due to an intrinsic limitation of this type of data, namely the High Feature-to-Sample Ratio in spectral data, which inherently increases the risk of overfitting in linear models.

Regarding the high performance of the Decision Tree (DT) model, we have added a detailed discussion (Lines 757–789) to explain this phenomenon. The added text highlights DT’s ability to capture non-linear relationships in complex spectral data, the positive impact of metaheuristic feature selection, cultivar-specific responses, and comparison with other algorithms (ANN, SVM-R). Relevant studies have also been cited to support these points (da Silva et al., 2025; Ren et al., 2020; Wang et al., 2015). We believe this addition clarifies the DT-related observations and addresses the reviewer’s concern fully.

Reviewer 2 Report

Comments and Suggestions for Authors

The authors have made all the necessary changes in the revised version of the manuscript. I have no more comments. The article can be accepted in its present form.

Author Response

Comment 1: The authors have made all the necessary changes in the revised version of the manuscript. I have no more comments. The article can be accepted in its present form.

Response 1: Thank you very much for your thorough review and positive feedback on our revised manuscript. We sincerely appreciate the time and effort you have dedicated to evaluating our work and providing constructive comments that have significantly improved the quality of our paper.